# Physical and Chemical Properties of Convective- and Microwave-Dried Blackberry Fruits Grown Using Organic Procedures

**DOI:** 10.3390/foods13050791

**Published:** 2024-03-04

**Authors:** Marko Petković, Nemanja Miletić, Valerija Pantelić, Vladimir Filipović, Biljana Lončar, Olga Mitrović

**Affiliations:** 1Department of Food Technology, Faculty of Agronomy Čačak, University of Kragujevac, Cara Dušana 34, 32000 Čačak, Serbia; n.m.miletic@kg.ac.rs (N.M.); p.valerija.19@gmail.com (V.P.); 2The Faculty of Technology Novi Sad, University of Novi Sad, Bulevar Cara Lazara 1, 21000 Novi Sad, Serbia; vladaf@uns.ac.rs (V.F.); cbiljana@uns.ac.rs (B.L.); 3Department for Fruit Processing Technology, Fruit Research Institute, Kralja Petra I 9, 32000 Čačak, Serbia; omitrovic@institut-cacak.org

**Keywords:** blackberry (*Rubus fruticosus*), convective drying, microwave drying, polyphenols, antioxidant capacity

## Abstract

This study aimed to evaluate the effect of convective and microwave drying on the bioactive-compounds content of blackberry (*Rubus fruticosus*) fruits, as well as drying parameters and energy consumption. The fruit was dehydrated in a convective dehydrator at a temperature of 50 °C and 70 °C and in a microwave oven at power levels of 90 W, 180 W and 240 W. The highest amount of anthocyanins, polyphenols and antioxidant capacity were obtained in blackberry fruits that were microwave dried at 90 W and 180 W (46.3–52.5 and 51.8–83.5 mg 100 g^−1^ dm of total anthocyanins, 296.3–255.8 and 418.4–502.2 mg 100 g^−1^ dm of total phenolics, and 1.20–1.51 and 1.45–2.35 mmol TE 100 g^−1^ dm of antioxidant capacity for 90 W and 180 W models, respectively). It turned out that microwave dehydration shortened the processing time and lowered the energy consumption compared to convective drying (a significantly reduced drying time of 92–99% with microwave dehydration). Blackberry fruits dehydrated at 240 W showed the shortest dehydration time (59–67 min), minimal energy consumption (0.23 kWh) and the most efficient diffusion (1.48–1.66 × 10^−8^ m^2^ s^−1^).

## 1. Introduction

Production of blackberry fruits (*Rubus fruticosus*) is constantly increasing worldwide. The advantages of cultivation are reflected in its early fruit-bearing, regular and high yields, and adaptation to different cultivation systems. Due to their rich biochemical composition, blackberries possess nutritious and medicinal properties, which makes them an important part of the human diet [1,2].

The demand for fresh, naturally preserved and quality products which are physically and chemically treated as little as possible during processing is increasing. On the other hand, blackberry fruits are extremely perishable and have a short market life, and thus various forms of processing and/or deep freezing are a necessity. Dried fruit, as a concentrated form of fresh fruit, is mainly consumed as a handheld snack due to its delicate organoleptic properties and high energy content. This image of dried fruits, including dried berry fruits, has recently been changing due to their high antioxidant capacity and beneficial influences on human health [3]. The drying process can significantly impair the sensory properties of fruits, and thus it is of essential importance to choose the proper dehydration method based on the properties of the fresh fruit and the desired properties of the dried products. The goal of every dehydration procedure is to reduce the negative changes in the raw material, to preserve the content of bioactive substances to the greatest extent possible and to thereby enable obtaining a quality product with an extended shelf life. Regarding energy consumption, the duration of the selected dehydration method should be shortened as much as possible [4].

Unlike convective drying, where moisture is initially removed from the surface while, subsequently, the remaining moisture diffuses from the inner parts towards the surface until complete dehydration, in microwave drying, the heat is generated directly in the interior of the material, creating a greater heat transfer and resulting in faster evaporation of moisture [5,6]. Microwave drying takes significantly less time compared to the convective procedure [7]. 

The aim of this work was to examine the possibility of applying convective and microwave dehydration to blackberry fruits of the Loch Ness and Triple Crown varieties grown using organic procedures. The drying parameters varied for both applied methods, while the energy, kinetic and control parameters were monitored. 

## 2. Materials and Methods

### 2.1. Fruit Sampling

Two thornless cultivars of blackberry fruits, Loch Ness and Triple Crown, were collected in July 2022 in a family orchard in the village Gornji Dubac, Serbia (43°39′54″ N 20°21′56″ E, altitude 850 m) and stored in sealed plastic bags at −18 °C for no longer than one month. Each sample consisted of approximately 400 blackberry samples: 10 berry fruits at the full maturity stage (harvest maturity) from 40 bushes, with no mechanical injuries, were randomly selected. Before the dehydration process, the frozen blackberries were washed with cold water and allowed to stabilize for a few hours at room temperature.

The production of these two blackberry varieties was carried out according to organic principles (under a certificate issued by the authorized organization Ecocert Balkan, Beograd, Serbia).

### 2.2. Drying Procedures

The Loch Ness and Triple Crown blackberry fruits were subjected to convective and microwave drying processes. Prior to drying, the fruits were visually selected according to size and color, and damaged and moldy samples were removed. Convective drying of the berry fruits was carried out in a commercial food dehydrator (Gorenje FDK 500GCW, Velenje, Slovenia) at the temperatures of 50 °C and 70 °C, an air flow speed of 7.9 ms^−1^ and atmospheric pressure, until constant mass. Five drying trays were parallelly placed into the drying chamber. Berry samples were positioned on the drying trays in a monolayer formation, while the heated air was introduced vertically across the trays from the bottom to the top. Each tray initially held approximately 100 g of berry fruits (1.325 kgm^−2^). The positions of the trays were changed every hour, in such a manner that the top tray was placed as the bottom one. 

Microwave drying was processed in a commercial microwave oven (Tesla MW2390MB 1250 W, Praha, Czech Republic), applying power of 90, 180 and 240 W. The berries (~100 g) were processed into a single mono-layer formation until constant mass. 

The consumption of energy and the amount of emitted CO_2_ in the atmosphere were monitored by the consumption meter (Prosto PM 001, Jiangbei District, Ningbo, China). The mathematical relation between energy consumption and oxidation was 1 kWh = 0.998 kg CO_2_. 

### 2.3. Extraction and Determination of Total Anthocyanins, Total Phenolics and Antioxidant Capacity

Dried blackberry samples (50 g) were powdered and mixed with 250 mL of 96% ethanol and ultrasonicated. After 30 min of extraction at 25 °C, the mixture was centrifuged two sequential times for 15 min at 3500 rpm, and the supernatant was filtered through a 0.45 mm Minisart filter before analysis. The obtained extracts were used to determine the total polyphenolic contents and antioxidant capacity. The quantification of the total phenolic content was conducted using a modified Folin–Ciocalteu colorimetric method, and the results were expressed as milligrams of gallic acid equivalents per 100 g of dry matter (mg GAE100 g^−1^ DM) [8,9]. Antioxidant properties, quantified as the Trolox equivalent100 g^−1^ dry matter (mmol TE100 g^−1^ DM), were determined by the ABTS assays [10]. An identical extraction procedure was repeated but with 25 mL of 96% ethanol/HCl (85:15 *v*/*v*) in order to obtain extract for anthocyanin content. The pH-differential method was employed to determine the monomeric anthocyanin pigment content in the fruit extracts, and the results were expressed as milligrams of cyanidin-3-glucoside equivalents/100 g dry matter (mg cyn-3-glu100 g^−1^ DM) [11]. All these determinations were performed in triplicate, and the results were presented as mean value of three measurements ± standard deviation.

### 2.4. Determination of Total Soluble Solids Content, Dry Matter Content, pH and Ash Content 

The soluble solid content of the fresh blackberries was determined on a manual refractometer (3828, Carl Zeiss, Jena, Germany), while dry matter content was determined by drying ~10 g of berries at 105 °C until constant mass. The ash content was measured for a 5 g sub-sample using a muffle furnace at 550 °C until constant mass.

### 2.5. Statistical Analysis

All assays were carried out in triplicate. Data were analyzed by one-way analysis of variance (ANOVA), using CoStat software 6.311. The paired comparisons between different parameters were performed using Tukey’s test (*p* < 0.05). To evaluate and distinguish all examined parameters of the blackberries (drying time, effective moisture diffusivity, energy consumption/emission of CO_2_, total anthocyanins, total phenolics and antioxidant activity), a covariance matrix was employed for comparison through Principal Component Analysis (PCA) [12]. Pearson correlation was computed, and a significance level of *p* < 0.05 was applied. The color correlation diagram between the derived mass transfer rate parameters, drying time, effective moisture diffusivity, energy consumption/emission of CO_2_, total anthocyanins, total phenolics and antioxidant activity was generated using the R Studio 1.4.1106 program [13].

## 3. Results and Discussion

### 3.1. Chemical Properties of Fresh and Dried Blackberry Fruits

The basic physico-chemical properties of the fresh blackberry samples were determined (Table 1). There were no significant statistical differences between the two varieties. The obtained results for dry matter content, total soluble content, acidity (pH) and mineral content were either slightly lower or comparable to the previously published data [14,15,16].

The chemical composition of the fresh and dried blackberry fruits was analyzed and is presented in Table 2. Fresh fruits of the Loch Ness variety contained significantly higher amounts of bioactive compounds, such as the total anthocyanins and total phenolics, compared to the fresh fruits of The Triple Crown variety. Consequently, a higher level of antioxidativity was shown by the Loch Ness samples. 

Any kind of dehydration process resulted in a loss in the content of bioactive compounds. Applying high temperatures certainly causes degradation, due to the high thermolability of polyphenolics, especially anthocyanins [17]. Regarding our results, one can conclude that the level of preservation of bioactive compounds is higher if a higher temperature was applied, which is related to the duration of convective drying (Table 2). Specifically, convective drying of Loch Ness fruits at 50 °C and 70 °C lasted 9629 min and 3086 min, respectively, while the drying of Triple Crown samples at the same temperatures lasted 9903 min and 3255 min, respectively. Although the sensitivity of polyphenolics increases with temperatures, the duration of the processes at 50 °C is ~3 times longer, which prompted degradation to a greater extent at this thermal level.

In microwave drying, the highest level of bioactive compounds and antioxidativity were achieved at 180 W, while the most dominant degradation occurred during drying at 240 W in both cultivars. Again, the duration of the dehydration process played an important role in the preservation of bioactive compounds, and, thus, the level of degradation was higher at the drying power of 90 W compared to 180 W. Namely, drying both blackberry varieties at 90 W and 180 W lasted 197 min and 71 min, respectively. A three-times prolonged drying process, albeit at half the power, led to a higher level of degradation. A comparison between the two drying methods led to the conclusion that microwave drying preserves the anthocyanins and polyphenols to a greater extent than convective drying. Such a conclusion is supported by previously published works [7,18]. Additionally, according to Stamenković et al., the ideal conditions for the convective drying of raspberries consist of a temperature set at 60 °C and a hot air speed of 1.5 ms^−1^ [19,20].

### 3.2. Thin-Layer Convective and Microwave Drying of Blackberries

The fresh Loch Ness blackberry fruits’ initial moisture content was 5.53 ± 0.31 kg H_2_O kg^−1^ DM, and the Triple Crown’s was 5.57 ± 0.31 kg H_2_O kg^−1^ DM [21]. During the dehydration of the fruits, the moisture ratio (*MR*) over time was followed, and the results are presented in Figure 1. *MR* is defined as the following ratio: [(*M*_t_ − *M*_e_)/(*M*_0_ − *M*_e_)], where *M*_0_ is initial moisture content, *M*_e_ is equilibrium moisture content and *M_t_* is the moisture content at a given time on the dry basis. With increasing temperature or power, the curves are steeper, which indicates a shorter time of fruit dehydration. If the drying temperature increases, the partial pressure of water vapor on the surface of the fruit also increases, resulting in a faster diffusion of moisture from the interior to the surface of the fruit. All dehydration curves had the same shape, with different drying times to constant mass. The drying process concluded once the dehydrated fruit reached a constant mass (*MR* ≈ 0, 0.18 kg H_2_O kg^−1^ db). The Loch Ness fruits had the longest drying time at a temperature of 50 °C (9629 min), while drying at a microwave power of 240 W took the least time (59 min, for both varieties). The drying curves of the Triple Crown fruits using the microwave method were steeper compared to the Loch Ness curves, and the drying processes were shorter. Irrespective of the dehydration technique employed, a swift reduction in water content was observed in the initial phase of the drying process. The dehydration time from the initial to the final moisture contents in Eminoğlu et al.’s results were measured as 2040, 1350, 1050 and 930 min for air-drying temperatures of 54, 61, 68 and 75 °C, respectively [22]. The drying experiments for blackberries using microwave and convective dehydration methods indicate a significantly reduced drying time of 92–99% with microwave dehydration. Similar results could be observed in the work of Pantelić, where the savings in microwave drying of raspberries were 86–96% [23].

The *MR* gauges the moisture level in a food item undergoing drying through microwave and convective energy. This parameter plays a crucial role in the drying process as it dictates both the speed at which moisture is extracted from the product and the ultimate moisture content. A decreased moisture ratio (*MR*) results in a decreased final moisture content, while an increased *MR* leads to a higher ultimate moisture content. Greater moisture content promotes water evaporation [24]. Typically, using a microwave with higher wattage and employing convective dehydration at a higher temperature range statistically significantly (*p* < 0.05) will result in a quicker dehydration rate compared to the slower rate achieved with a lower-wattage microwave (Table 3, Figure 1).

The drying rate (*DR*) represents the total mass loss of dehydrated materials (*M*_i−1_ − *M*_i_) between the two consecutive measurements (*t*_i−1_ − *t*_i_) on a defined tray [DR = (*M*_i−1_ − *M*_i_)/(*t*_i−1_ − *t*_i_)]. As the temperature and power increased, the *DR* also increased. The highest *DR* was achieved at equivalent dehydration durations for both examined blackberry fruits, employing an identical dehydration model (Figure 2). The maximum value of *DR* for convective drying was achieved after 960 min of drying at a temperature of 50 degrees (*DR*_max_ = 0.019 g min^−1^ for LN and *DR*_max_ = 0.017 g min^−1^ for TC), 480 min of drying at a temperature of 70 °C (*DR_max_* = 0.053 g min^−1^ for LN and *DR*_max_ = 0.048 g min^−1^ for TC), 30 min of drying at a microwave power of 90 W (*DR*_max_ = 0.651 g min^−1^ for LN and *DR*_max_ = 0.891 g min^−1^ for TC), and 20 min of drying at a microwave power of 180 W (*DR*_max_ = 2.049 g min^−1^ for LN and *DR*_max_ = 1.916 g min^−1^ for TC) and 240 W (*DR*_max_ = 2.261 g min^−1^ for LN and *DR*_max_ = 2.456 g min^−1^ for TC). The similarity in results between microwave drying at power levels of 180 and 240 W indicates that the fluctuation in power did not have a notable impact on the drying outcome. With an increase in temperature and microwave power, the *DR*_max_ also increases. It can be concluded that the drying time, as well as the values of *MR* and *DR*, will depend statistically more (*p* < 0.05) significantly on the chosen drying method (convective or microwave) and its parameters (temperature, power range), regardless of the cultivars. A similar finding was found in Lackowicz’s results [18]. The minimum *DR* ratio of blackberries dehydration was increased 12–119 times with microwave dehydration. Such an effective influence of microwave energy was found in the work of Pantelić, where the *DR*_max_ was increased up to 19 times [23]. The convective drying method of raspberries was found to be quite prolonged, with a total drying duration of 1126 ± 8 min. However, the microwave-convective drying trial, where microwaves were consistently used throughout the process (with air temperature set at 55 °C, air velocity at 2 m/s and microwave power at 100 W), unfortunately posed a risk of overheating and causing burning effects on the sample [24].

### 3.3. Determination of Effective Moisture Diffusivity and Energy of Activation

The effective moisture diffusivity *D*_eff_ can be determined through the application of Fick’s second law of diffusion, considering the fruit’s spherical shape (Equation (1)) [18]:(1)$$\mathrm{MR}=\frac{6}{\pi^{2}}\;\times \;\sum_{i=1}^{\infty }\frac{1}{J_{0}^{2}}\times \;e^{-\frac{J_{0}^{2}\;\times \;D_{\mathrm{eff}}}{{4\;\times \;r}^{2}}}$$

*D*_eff_ is the effective moisture diffusivity (m^2^ s^−1^), *t* is time (s), *J*_0_ is the roots of the Bessel function and *r* is the blackberries’ radius (sphere is the appropriate model for the berries). If the *D*_eff_ was constant in a relatively long drying period, Equation (1) could be transformed in ln(*MR*) = ln(*a*) − *k* × *t*. The linear relation ln(*MR*) and t gives the possibility of calculating the equation slope, which is equal to the drying constant (*k*, Equation (2)):(2)$$k=-\frac{\pi^{2\;}\times D_{\mathrm{eff}}}{{4\;\times \;r}^{2}}$$

An Arrhenius equation, Equation (3) for convective drying and Equation (4) for microwave drying, could be used for the energy of activation calculation, *E*_a_ [25]:(3)$$D_{\mathrm{eff}}=D_{0}\;{\times \;e}^{-\frac{E_{a}}{\mathrm{RT}}}$$
(4)$$D_{\mathrm{eff}}=D_{0}{\times \;e}^{-\frac{E_{a}\;\times \;m}{P}}$$

*E*_a_ (kJ mol^–1^) is the energy of activation, *R* (8.3143 J mol^–1^ K^–1^) is the universal gas constant, *T* (K) is the absolute air temperature and *D*_0_ (m^2^ s^–1^) is the pre-exponential factor of the Arrhenius equation. The Equations (3) and (4) could be transformed into the linear equations:(5)$$\ln(D_{\mathrm{eff}})=\ln(D0)\;-\;k\;\times \;{\left(T+273.15\right)}^{-1}$$
(6)$$\ln(D_{\mathrm{eff}})=\ln(D0)\;-k\;\times \;mP^{-1}$$

The linear relation ln(*D*_eff_) and *T* gives the possibility of calculating the equation slope, which is equal to the drying constant *k* = *E*_a_ × *R*^−1^. The natural logarithm of *D*_eff_ versus mass load *m* (g)/*P* (W) was used to calculate the *E*_a_ (W g^−1^) of microwave drying.

Elevated air-drying temperatures and higher microwave power statistically significantly (*p* < 0.05) resulted in higher *D*_eff_ values due to enhanced moisture diffusion at elevated temperatures (Table 3). The highest *D*_eff_ values were calculated for the experimental microwave drying at 240 W (1.48 × 10^−8^ ± 1.10 × 10^−9^ m^2^ s^−1^ for Loch Ness and 1.66 × 10^−8^ ± 9.48 × 10^−10^ m^2^ s^−1^ for Triple Crown). The presented *D*_eff_ values were within the specific ranges (10^−8^–10^−11^ m^2^ s^−1^), according to the previous published data [26,27,28]. The moisture diffusion and *D*_eff_ were not dependent on the blackberry diameter since the diameter of Loch Ness was 23.76 ± 0.75 mm and 24.66 ± 0.81 mm for Triple Crown. It was also observed that the *D*_eff_ values for chokeberry dehydration were highest when using the maximum microwave power and the highest convective drying temperature [29]. The moisture diffusivity of drying *Curcuma longa* L. slices ranged from 0.85 × 10^–8^ m^2^ s^−1^ to 2.15 × 10^–8^ m^2^ s^−1^ [30].

The energy of activation *E*_a_ reflects the sensitivity of diffusivity to temperature and power range, indicating the energy required to initiate water diffusion. A higher *E*_a_ signifies increased sensitivity of *D*_eff_ to changes in temperature and power. *E*_a_, for the convective drying, was calculated to be 54.45 kJ mol^−1^ for both varieties, while for microwave drying was 16.66 ± 1.63 W g^−1^ for Loch Ness and 12.06 ± 0.71 W g^−1^ for Triple Crown. As could be concluded for *D*_eff_, *E*_a_ was not depended on the blackberry diameter but on the type of drying and its parameters. A reduced *E*_a_ implies enhanced effective moisture diffusivity (higher *D*_eff_) and increased moisture diffusion with the radius (thickness) of the sphere. This suggests that lower energy consumption leads to the breaking of bonds between water molecules in the sample [28] and was correlated with the results presented in the work of Pantelić [23], where the diameter of raspberries was about 25 mm (*E*_a_ = 65.22 kJ mol^−1^). The calculated *E*_a_ was within the specific ranges (12.7 to 110 kJ mol^−1^), according with the previous research by Eminoğlu’s et al., in which the energy of activation was calculated as 42.25 kJ mol^−1^ [22]. Moreover, in the context of microwave drying of onion slices at different microwave output powers (328, 447 and 557 W), the Arrhenius equation was used to characterize the impact of temperature on diffusivity, with an activation energy of 45.60 kJ mol^−1^ [26]. The activation energy for the thin layer convective and microwave drying of hazelnuts (conducted at three temperatures of 40, 50 and 60 °C, and microwave powers of 0, 450 and 900 W) was between 15.61675 and 41.0053 kJ/mol [31].

### 3.4. Determination of Energy Consumption

Evaluating the energy usage I, as well as the emission of CO_2_, in dehydration procedures is crucial for gauging process efficiency and pinpointing potential energy-saving avenues. One approach to this assessment involves direct measurement, wherein the energy consumption analysis of the dehydration process entails directly measuring the energy input into the system. Microwave drying has become increasingly popular recently, offering improved alignment between energy usage and product quality. The experimental findings in Table 3 revealed a significant influence of the dehydration process on energy consumption (*E*), directly correlating with the duration of the drying process. It was evident that there was a statistically significantly reduction in energy input (*p* < 0.05) as the microwave energy for drying increased, accompanied by a subsequent decrease in drying time. The microwave drying at 180 and 240 W was the least energetically demanding (*E* = 0.19–0.23 kWh), while the convective drying was, at about 25–35 kWh, the more energetically demanding drying process. The results for microwave drying at 180 and 240 W power were almost the same, indicating that the variation in power had no significant effect on the outcome of the drying process. Additionally, the experimental findings indicated that convective drying of Loch Ness blackberries demanded a higher energy input compared to Triple Crown drying, whereas the opposite trend was observed for microwave drying. Microwave drying at power inputs of 180 and 240 W exhibited the shortest drying time and the highest *D*_eff_, influenced by the lowest energy demand in the drying process. The previous results for drying organic blackberries showed that increasing the microwave power decreased the amount of energy consumption quadratically [32]. Also, the use of pretreatment (e.g., ultrasound assistance) can significantly reduce the energy required for the convective drying of blackberries [33,34]. In previous research into different drying methods for raspberry, convective drying was identified as the most energy-intensive process (~0.9 MJ g^−1^), while microwave-assisted drying exhibited the lowest energy consumption (~0.1 MJ g^−1^) per gram of evaporated moisture [35]. The highest energy consumption for strawberries was for traditional convective drying (9.05 ± 0.22 kWh), while the lowest energy consumption was for microwave-supported drying (0.98 ± 0.06 kWh) [36].

### 3.5. Statistical Analysis

The drying method (convective and microwave drying) and their parameters (temperature and microwave power) were used as independent variables, and PCA was applied to identify the structure in the correlation between these parameters and dependent variables, the drying time, effective moisture diffusivity, energy consumption/emission of CO_2_, total anthocyanins and phenolics, and antioxidant activity. The results of the PCA are shown in Figure 3. A scatter plot was generated using the initial two principal components derived from the PCA of the data matrix. The first principal component was assigned to the x-axis, while the second principal component was assigned to the y-axis. The purpose was to illustrate patterns within the presented data and to showcase the efficacy of the descriptors employed in distinguishing between different data points. The angles between corresponding variables reflect the extent of their correlations, with smaller angles indicating stronger correlations [37,38]. A scatter plot was designed with the first two principal components (F1, F2) from the PCA data matrix. The first two components demonstrated 87.12% of the total variance in the experimental data. The contribution of the variables (%) showed that all variables except the effective moisture diffusivity (drying time, energy consumption/emission of CO_2_, total anthocyanins and phenolics, and antioxidant activity) equally participated in F1 (13.06–16.19%). The drying time and energy consumption/emission of CO_2_ most participated in F2 (22.60% and 16.07%, respectively). The position of the samples in Figure 3 was primarily more influenced by the type of drying than by the drying parameters and blackberry species. Similar correlations have been previously published [17,39]. Blackberries, regardless of their species, dried by microwave power were characterized by higher values of all analyzed parameters except energy consumption/emission of CO_2_ (oriented on the positive side of the x-axis by the positive value of the F1 component), while the convection-dried blackberries were oriented on the negative side of the x-axis (by the negative value of the F1 component). Therefore, the Triple Crown fruits dried using microwave power 180 W were characterized by high values of the following parameters: effective moisture diffusivity, total anthocyanins, total phenols and antioxidant activity. The blackberries dried by the convection method were characterized by higher energy consumption/emission of CO_2_. The high concentrations of anthocyanins and phenols contributed to the higher antioxidant activity of dried blackberries as well.

The microwave drying proved to be more effective in terms of drying time, with a statistically significantly shorter drying process time. The *D*_eff_ generally rises as temperature and energy input increase. This is attributed to the heightened mobility of moisture molecules at higher temperatures and energy input, leading to a faster diffusion rate. The findings presented in this paper affirm this observation, as the *D*_eff_ values were found to be highest in the microwave drying associated with the highest microwave power range. As energy input escalates, there is a notable increase in the *D*_eff_. The experimental findings reveal a notable influence of drying process duration on energy consumption. In particular, the results indicate a statistically significant decrease in energy input with an input of microwave energy and a subsequent reduction in drying time. This implies that substantial energy savings can be achieved by shortening the drying process duration. It is crucial to acknowledge that various drying processes may exhibit diverse energy requirements. Additionally, this discussion underscores that microwave drying proved to be less energetically demanding for the drying of blackberries. This suggests that selecting an appropriate drying method tailored to a specific product can contribute to reducing overall energy consumption.

A visual representatiIn in the form of a color correlation diagram (Figure 4) was generated to illustrate the statistical significance of the correlation coefficients between various variables and their corresponding responses (total anthocyanins and phenolics, antioxidative activity, drying time, effective moisture diffusivity, energy of activation and energy consumption/emission of CO_2_). The graphical display employs circle size and color (blue indicating positive correlation and red indicating negative correlation) to represent the values of the correlation coefficients among the tested parameters (Figure 4) [40]. A high level of positive correlation was shown between drying time and energy consumption/emission of CO_2_ (*r* = 0.8908, statistically significant at *p* < 0.05) and total anthocyanins—total phenolics—antioxidative activity (*r* = 0.9040–0.9167, statistically significant at *p* < 0.05). These results were expected since there is a direct dependence between the drying model (convective, microwave), drying time and total energy input. Also, antioxidative activity is directly dependent on the total anthocyanins and total phenolics content, as bioactive components.

A high level of negative correlation was shown between drying time/energy consumption (emission of CO_2_) and antioxidant activity (r = −0.7452 to −0.7597, statistically significant at *p* < 0.05). A lower level of negative correlation was found between drying time/energy consumption (emission of CO_2_) and total anthocyanins and total phenolics content (r = −0.6307 to −0.7290, statistically significant at *p* < 0.05). The bioactive components (anthocyanins, phenolics) are thermolabile, and their antioxidative activity will be decreased at a higher temperature or microwave power input, or longer dehydration time. The experimental results of this work are in correlation with previously presented results [32,41]. The results have implications for enhancing the drying process, particularly in sectors such as the food industry, where dehydration plays a crucial role in fruit processing and preservation.

## 4. Conclusions

This study successfully assessed the impact of convective and microwave drying on the bioactive compound content of organically grown blackberries (*Rubus fruticosus*). By employing convective drying at temperatures of 50 °C and 70 °C, as well as microwave drying at power levels of 90 W, 180 W and 240 W, we determined that the highest concentrations of anthocyanins, polyphenols and antioxidant capacity were achieved in microwave-dried blackberry fruits at 90 W and 180 W. Importantly, microwave drying demonstrated notable advantages, including reduced processing time and lower energy consumption when compared to convective drying methods. Particularly, blackberry fruits dehydrated at 240 W exhibited the shortest dehydration time, minimal energy consumption and the most efficient diffusion. These findings highlight the potential of microwave drying as a more time-efficient and energy-saving alternative for preserving bioactive compounds in blackberries. The insights gained from this study offer valuable information for developing innovative and efficient drying methods tailored to enhance the preservation of bioactive compounds in blackberries, further contributing to the advancement of food processing technologies. For example, combined convective and microwave drying can further improve (organic) blackberry drying methods.

## Figures and Tables

**Figure 1 foods-13-00791-f001:**
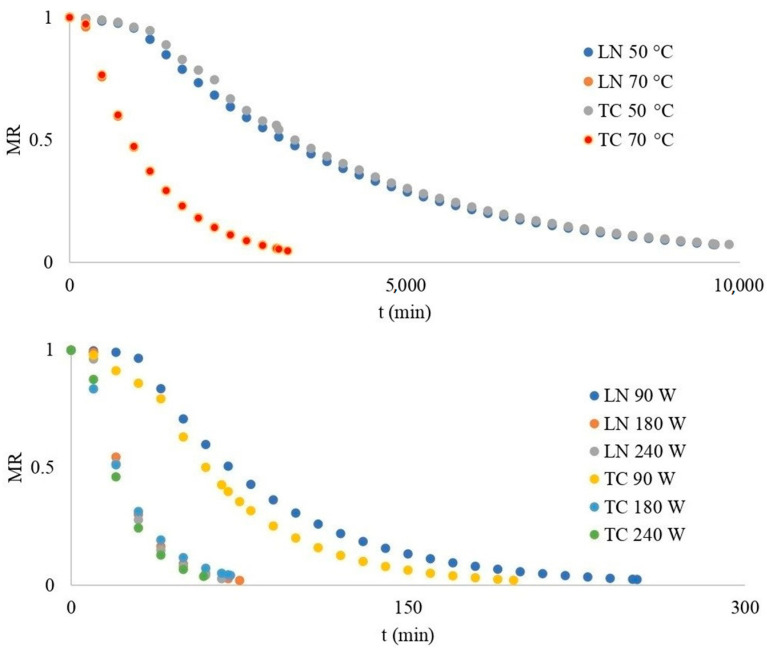
*MR* curves of convective drying (**upper** figure) and microwave drying (**lower** figure). LN and TC stand for Loch Ness and Triple Crown varieties.

**Figure 2 foods-13-00791-f002:**
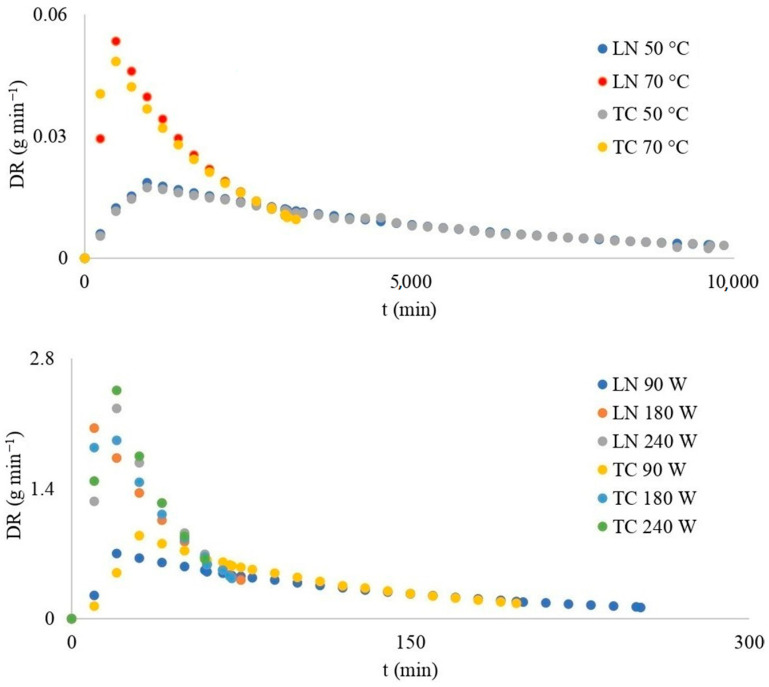
*DR* curves of convective drying (**upper** figure) and microwave drying (**lower** figure).

**Figure 3 foods-13-00791-f003:**
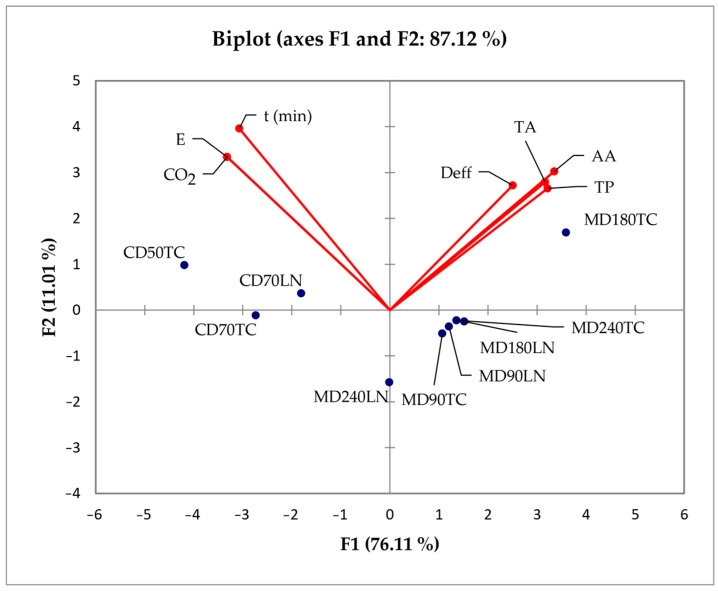
PCA of independent variables and responses of the convective and microwave drying.

**Figure 4 foods-13-00791-f004:**
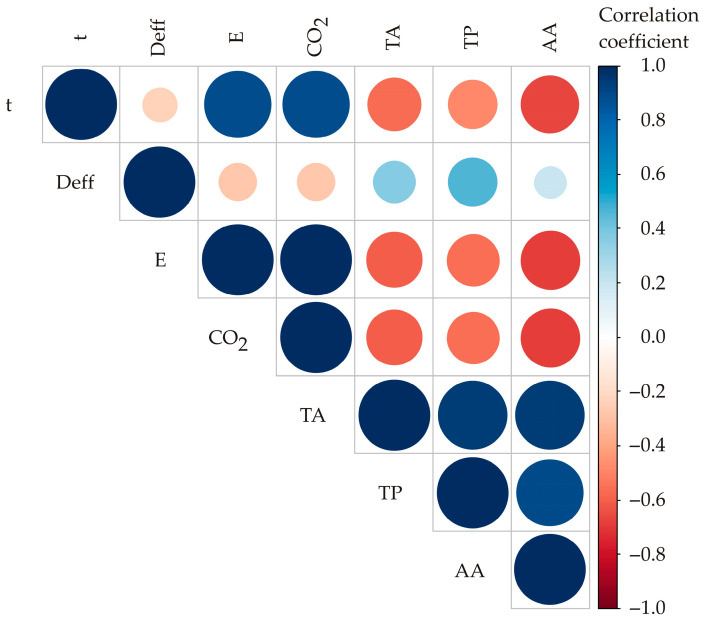
A color-correlation diagram depicting the relationship between the independent variable parameters and the responses of the convective and microwave drying.

**Table 1 foods-13-00791-t001:** The physico-chemical parameters of fresh blackberry fruits.

Cultivar	Dry Matter Content (%)	Soluble Solid Content (%)	pH	Ash Content (%)
Loch Ness	15.3 ± 0.1	9.7 ± 0.1	3.40 ± 0.15	0.34 ± 0.04
Triple Crown	15.2 ± 0.1	9.8 ± 0.1	3.61 ± 0.10	0.30 ± 0.05
ANOVA	ns	ns	ns	ns

There are no statistically significant differences between two varieties (Tukey’s test, *p* < 0.05). ns: not significant.

**Table 2 foods-13-00791-t002:** Chemical composition of fresh and dried blackberries.

		Total Anthocyanins (mg 100 g^−1^ DM)		Total Phenolics (mg 100 g^−1^ DM)		Antioxidant Capacity (mmol TE 100 g^−1^ DM)	
		Loch Ness	Triple Crown	Loch Ness	Triple Crown	Loch Ness	Triple Crown
Fresh		503.9 ±16.7 ^a^	331.0 ± 9.4 ^b^	1280.0 ± 150.5 ^a^	796.0 ± 151.7 ^b^	7.49 ± 0.94 ^a^	4.86 ± 0.84 ^b^
CD *	50 °C	1.3 ±0.2 ^g^	0.9 ±0.1 ^g^	149.8 ± 18.0 ^h^	79.3 ±3.0 ^i^	0.64 ± 0.03 ^fg^	0.43 ± 0.08 ^g^
	70 °C	16.7 ± 1.7 ^ef^	5.8 ± 0.3 ^fg^	229.6 ± 0.6 ^f^	53.1 ± 6.2 ^i^	0.95 ± 0.00 ^ef^	0.82 ± 0.07 ^efg^
MD *	90 W	46.3 ± 1.9 ^d^	52.5 ± 2.5 ^d^	296.3 ± 25.7 ^e^	255.8 ±0.1 ^f^	1.51 ± 0.13 ^d^	1.20 ± 0.09 ^de^
	180 W	51.8 ± 1.6 ^d^	83.5 ± 4.4 ^c^	418.4 ± 6.6 ^d^	502.2 ± 25.7 ^c^	1.45 ± 0.06 ^d^	2.35 ± 0.19 ^c^
	240 W	17.2 ± 1.4 ^ef^	19.9 ± 1.5 ^e^	196.0 ± 19.4 ^g^	246.1 ± 2.4 ^f^	0.95 ± 0.01 ^ef^	1.48 ± 0.10 ^d^
ANOVA		***		***		***	

* CD and MD stand for conductive drying and microwave drying, respectively. ** Values with different letters within the analyzed trait (total anthocyanins, total phenolics, antioxidant capacity) and both varieties (Loch Ness, Triple Crown) denote statistically significant differences (Tukey’s test, *p* < 0.05). ns, *, **, ***: not significant or significant at *p* < 0.05, 0.01, 0.001, respectively.

**Table 3 foods-13-00791-t003:** Drying parameters, energy usage and CO_2_ emission of CD and MD.

		Drying Time (min)		*D*_eff_ (m^2^ s^−1^)		*E*_a_ **		*E* (kWh)		CO_2_ (kg)	
		Loch Ness	Triple Crown	Loch Ness	Triple Crown	Loch Ness	Triple Crown	Loch Ness	Triple Crown	Loch Ness	Triple Crown
CD *	50 °C	9629 ± 41 ^b^	10,156 ± 94 ^a^	7.09 × 10^−11^ ± 6.02 × 10^−12 e^	7.77 × 10^−11^ ± 5.83 × 10^−12 e^	54.45 ± 2.54 ^a^	54.45 ± 1.94 ^a^	6.75 ± 0.21 ^b^	7.36± 0.26 ^a^	6.74 ± 0.30 ^b^	7.34 ± 0.26 ^a^
	70 °C	3086 ± 37 ^d^	3255 ± 47 ^c^	2.36 × 10^−10^ ± 2.32 × 10^−11 e^	2.59 × 10^−10^ ± 2.16 × 10^−11 e^			5.61 ± 0.28 ^d^	6.11 ± 0.18 ^c^	5.59 ± 0.28 ^d^	6.1 ±0.18 ^c^
MD *	90 W	252 ± 13 ^e^	197 ± 11 ^e^	3.98 × 10^−9^ ± 2.94 × 10^−10 d^	5.94 × 10^−9^ ± 1.88 × 10^−10 c^	16.66 ±1.63 ^a,^*	12.06 ± 0.71 ^a,^*	0.38 ± 0.02 ^e^	0.30 ±0.02 ^e^	0.38 ± 0.02 ^e^	0.29 ± 0.02 ^e^
	180 W	75 ± 8 ^f^	71 ± 7 ^f^	1.42 × 10^−8^ ± 6.86 × 10^−10 b^	1.43 × 10^−8^ ± 4.05 × 10^−10 b^			0.23 ± 0.01 ^e^	0.19 ±0.01 ^e^	0.23 ± 0.01 ^e^	0.19 ± 0.02 ^e^
	240 W	67 ± 7 ^f^	59 ± 5 ^f^	1.48 × 10^−8^ ± 1.10 × 10^−9 b^	1.66 × 10^−8^ ± 9.48 × 10^−10 a^			0.23 ± 0.02 ^e^	0.21 ± 0.01 ^e^	0.23 ± 0.02 ^e^	0.21 ± 0.01 ^e^
ANOVA		***		***		***		***		***	

* CD and MD stand for conductive drying and microwave drying, respectively. ** *E*_a_: CD, kJ mol^−1^; MD, Wg^−1^. *** Values with different letters within trait (drying time, *D*_eff_, energy consumption, CO_2_ emission) and both varieties (Loch Ness, Triple Crown) denote statistically significant differences (Tukey’s test, *p* < 0.05). ns, *, **, ***: not significant or significant at *p* < 0.05, 0.01, 0.001, respectively. Activation energy (*E*_a_) was separately compared for conductive drying and microwave drying for both varieties due to the different energy units (kJ mol^−1^ and MD, Wg^−1^, respectively).

## Data Availability

The original contributions presented in the study are included in the article, further inquiries can be directed to the corresponding author.

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
