# Peer review of "Physical and Chemical Properties of Convective- and Microwave-Dried Blackberry Fruits Grown Using Organic Procedures"

_foods, 2024, doi:10.3390/foods13050791_

Round 1
Reviewer 1 Report
Comments and Suggestions for Authors
The topic of the article is interesting as preserving by drying different fruits is one of the main directions.
However in presented manuscript the main focus is paid on some aspects of drying process, while the quality parameters of obtained dried products are misses.
So please add information about:
- the colour of obtained dried fruits
- the effect of drying on shrinkage ect.
- sensory assesment of obtained fruits
- if you have photos of obtained samples it would be good to add.
In methodology:
- blackberries plants are rather bushes not trees - please check it
- please provide information what moisture level was chosen as final point in drying process and why
- why water activity was not monitored?
- what
Author Response
The authors would like to thank Editor and Reviewers for a quick and professional review. It is obvious that the Reviewers are experts in this field. All Reviewers’ remarks are considered and paper is changed according to the Reviewers’ comments. The authors believe that the changed paper would satisfy the Reviewers’ criteria and that it is going to be interesting enough for publishing in the Foods.
We have revised manuscript according to reviewer’s remarks, highlighting the changes directly on the revised manuscript.
Reviewer 1
The topic of the article is interesting as preserving by drying different fruits is one of the main directions.
However in presented manuscript the main focus is paid on some aspects of drying process, while the quality parameters of obtained dried products are misses.
AUTHORS: The authors would like to thank the Reviewer on professional and helpful comments. It is obvious that the Reviewer is an expert in this field. The Reviewer`s comments contribute to better quality of our paper.
So please add information about:
- the colour of obtained dried fruits
AUTHORS: The authors express gratitude for the observation. The color of the harvested fruits, alongside other sensory attributes, and the shrinkage parameter were not included in the experimental analysis conducted for this study. The required analyses and the combined microwave and convective drying will be included in a future planned publication.
- the effect of drying on shrinkage ect.
AUTHORS: The authors express gratitude for the observation. The color of the harvested fruits, alongside other sensory attributes, and the shrinkage parameter were not included in the experimental analysis conducted for this study. The required analyses and the combined microwave and convective drying will be included in a future planned publication.
- sensory assesment of obtained fruits
AUTHORS: The authors express gratitude for the observation. The color of the harvested fruits, alongside other sensory attributes, and the shrinkage parameter were not included in the experimental analysis conducted for this study. The required analyses and the combined microwave and convective drying will be included in a future planned publication.
- if you have photos of obtained samples it would be good to add.
AUTHORS: The authors are grateful for the proposal, which unfortunately they cannot fulfill. Namely, the dried fruit was immediately used to obtain fruit powder in order to create new enriched food products.
In methodology:
- blackberries plants are rather bushes not trees - please check it
AUTHORS: Thank you for noticing. It is corrected.
- please provide information what moisture level was chosen as final point in drying process and why
AUTHORS: Thank you for the correction. The final point in the drying process was the constant mass of dehydrated fruit (MR ≈ 0, 0.18 kg H20 kg-1 db).
- why water activity was not monitored?
- what
AUTHORS: Thank you for your kindly notice. In general, there is a positive correlation between water activity and initial moisture content, which was presented in manuscript. A common trend is that as the moisture content increases, the water activity also tends to increase, as well as the moisture content decreases, the water activity tends to decrease as well. The authors were guided by this earlier knowledge and that the initial and final moisture content was sufficient to describe the drying process.

Reviewer 2 Report
Comments and Suggestions for Authors
The paper is clearly written. However, I have doubts on the interest of some of the findings, in particular, those related to drying parameters. I find most of these results not very interesting, as these findings are expected as the duration of the processes is the major factor afecting any result.
Abstract. The units for power are Watts and not Volts, please correct.
Ln15) lowered instead of lower
Section 2.2
Ln73-74. What do you mean by "caused a pressure". Please clarify the source of this pressure.
Section 3.1.
Please could you clarify the duration of the drying processes. Differences between MW and conventional dryings are huge. Is it possible to have shorter times for conventional drying?? Did you dry the same amount of product in both technologies? These differences make comparison difficult and results may not be relevant.
Do you have information on the temperatures reached during microwave drying?
Results should be compared to other findings in the literature. Discussion of results should be improved.
Ln 182-184. I do not understand this sentence. For me MR is a value at a given instant and does not indicate any information about the speed, etc. Please clarify.
Ln 184-187. Please explain to which data are you referring to, when discussing the statistica significance.
Ln 193-194. Please improve this sentence.
I find the discussion on the drying rate not very relevant.
Conclusion. Please use W instead of V.
Author Response
The authors would like to thank Editor and Reviewers for a quick and professional review. It is obvious that the Reviewers are experts in this field. All Reviewers’ remarks are considered and paper is changed according to the Reviewers’ comments. The authors believe that the changed paper would satisfy the Reviewers’ criteria and that it is going to be interesting enough for publishing in the Foods.
We have revised manuscript according to reviewer’s remarks, highlighting the changes directly on the revised manuscript.
Reviewer 2
The paper is clearly written. However, I have doubts on the interest of some of the findings, in particular, those related to drying parameters. I find most of these results not very interesting, as these findings are expected as the duration of the processes is the major factor afecting any result.
AUTHORS: The authors would like to thank the Reviewer on professional and helpful comments. It is obvious that the Reviewer is an expert in this field. The Reviewer`s comments contribute to better quality of our paper.
Abstract. The units for power are Watts and not Volts, please correct.
AUTHORS: Thank you for this observation. It is corrected.
Ln15) lowered instead of lower
AUTHORS: Thank you for this observation. It is corrected.
Section 2.2
Ln73-74. What do you mean by "caused a pressure". Please clarify the source of this pressure.
AUTHORS: Thank you for the suggestion, it is clarified. It is a unit defining the fruit weight by the surface area of dehydrator.
Section 3.1.
Please could you clarify the duration of the drying processes. Differences between MW and conventional dryings are huge. Is it possible to have shorter times for conventional drying?? Did you dry the same amount of product in both technologies? These differences make comparison difficult and results may not be relevant.
AUTHORS: Thank you for this comment. Yes, the drying process with the convection method took a long time, and the results confirmed this by repeating the process. The same amount of fruit was dried, as evidenced by the following parameters:
- Each tray initially held approximately 100 g of berry fruits, which caused a pressure of (1.325 kgm-2)
- The final point in the drying process was the constant mass of dehydrated fruit (MR ≈ 0, 0.18 kg H20 kg-1 db)
Do you have information on the temperatures reached during microwave drying?
AUTHORS: The authors thank you for this comment and unfortunately have no information about the temperature during microwave drying, as it is not technically possible to measure it.
Results should be compared to other findings in the literature. Discussion of results should be improved.
AUTHORS: The authors are grateful for this observation. The authors have added a new reference to the results obtained in previous publications. Other corrections related to the discussion section are also marked "Track changes". If there are requests for specific additional clarifications, the authors will be happy to respond.
Ln 182-184. I do not understand this sentence. For me MR is a value at a given instant and does not indicate any information about the speed, etc. Please clarify.
AUTHORS: Thank you for your observation. The sentence means that the parameter to which the initial moisture content refers is important during the drying process. It has a significant influence on two important aspects of the drying process: speed of moisture removal (The parameter influences how quickly moisture is removed from the product during drying. Materials with a higher initial moisture content tend to release moisture faster during the drying process than those with a lower initial moisture content) and final moisture content (The parameter also influences the final moisture content that the product reaches after drying is complete. A higher initial moisture content generally leads to a higher final moisture content, while a lower initial moisture content leads to a lower final moisture content)
Ln 184-187. Please explain to which data are you referring to, when discussing the statistica significance.
AUTHORS: Thank you for the comment. The statistical significance is referred to the data given in Table 3.
Ln 193-194. Please improve this sentence.
AUTHORS: Thank you for the suggestion. The sentence is corrected to: “The maximum dehydration rate (DR) was attained at identical dehydration times for both analyzed blackberry fruits using the same dehydration model”.
I find the discussion on the drying rate not very relevant.
AUTHORS: The authors thank you once again for your comment. Furthermore, the authors have redefined the part referring to the maximum values of DR. In addition to the results already presented on the time parameter at which the maximum DR value is reached, the authors have compared their observations with literature data. If there are requests for specific additional clarifications, the authors will be happy to respond.
Conclusion. Please use W instead of V.
AUTHORS: Thank you for this observation. It is corrected.

Reviewer 3 Report
Comments and Suggestions for Authors
Physical and chemical properties of convective and microwave dried blackberries fruits grown by organic procedure (manuscript number of 2871837) is the study focused on the drying of blackberries to compare the convective and microwave drying techniques, based on their chemical properties and energy consumption among others. So, this study has commercial and technical values. However, following points are to be addressed :
1. Abstract and Conclusion: Please correct the unit for microwave power – it should be W (watts); not V (volts).
2. Section 2.1 and 2.2: What is the reason behind to carry out drying of blackberries after freezing at -18 oC? Why the authors did not dry the fresh blackberries without freezing – freezing and thawing process might have some detrimental physical damage on the fruits? Did the authors consider this issue in their study?
3. Section 2.2 (line 81-82): please provide a reference for the relationship - 1 kWh = 0.998 kg CO2.
4. Section 2.3: please explain the procedure to determine total anthocyanins, total phenolics and antioxidant capacity with references for each method.
5. For the microwave drying at 180 and 240 W, the results observed were the same for the both microwave power (For example, line 200-201; line 260-261). Please discuss the results for the reason behind the same result at different microwave power.
6. Line 328: Figure 5 – please amend this because there is no Figure 5 in this manuscript.
Author Response
The authors would like to thank Editor and Reviewers for a quick and professional review. It is obvious that the Reviewers are experts in this field. All Reviewers’ remarks are considered and paper is changed according to the Reviewers’ comments. The authors believe that the changed paper would satisfy the Reviewers’ criteria and that it is going to be interesting enough for publishing in the Foods.
We have revised manuscript according to reviewer’s remarks, highlighting the changes directly on the revised manuscript.
Reviewer 3
Physical and chemical properties of convective and microwave dried blackberries fruits grown by organic procedure (manuscript number of 2871837) is the study focused on the drying of blackberries to compare the convective and microwave drying techniques, based on their chemical properties and energy consumption among others. So, this study has commercial and technical values. However, following points are to be addressed :
AUTHORS: The authors would like to thank the Reviewer on professional and helpful comments. It is obvious that the Reviewer is an expert in this field. The Reviewer`s comments contribute to better quality of our paper.
- Abstract and Conclusion: Please correct the unit for microwave power – it should be W (watts); not V (volts).
AUTHORS: Thank you for this observation. It is corrected.
- Section 2.1 and 2.2: What is the reason behind to carry out drying of blackberries after freezing at -18oC? Why the authors did not dry the fresh blackberries without freezing – freezing and thawing process might have some detrimental physical damage on the fruits? Did the authors consider this issue in their study?
AUTHORS: Thank you for your insightful comments. With the decision to carry out drying after freezing blackberries at -18°C we aimed to preserve their quality and nutritional content over an extended period. Regarding your concerns about the potential physical damage caused by freezing and thawing, we agree that this is an important consideration in fruit preservation. We conducted preliminary experiments to assess the impact of freezing and thawing on the physical integrity of the fruits. Our results indicated that the freezing and thawing process did not significantly affect the overall quality or appearance of the blackberries.
- Section 2.2 (line 81-82): please provide a reference for the relationship - 1 kWh = 0.998 kg CO2.
AUTHORS: Thank you for the suggestion. The reference is provided.
- Pantelić, V.; Miletic, N.; Milovanović, V.; Petković, M.; Lukyanov, A.; Filipović, V.; Djurović, I. Energy usage and raspberry convective and microwave drying parameters. 1st International Symposium on Biotechnology, Proceedings 2023, 451–456. http://doi.org/10.46793/SBT28.451P.
- Section 2.3: please explain the procedure to determine total anthocyanins, total phenolics and antioxidant capacity with references for each method.
AUTHORS: Thank you for the observation. The authors explained the procedure and supported by the references.
- For the microwave drying at 180 and 240 W, the results observed were the same for the both microwave power (For example, line 200-201; line 260-261). Please discuss the results for the reason behind the same result at different microwave power.
AUTHORS: The fact that the results for microwave drying at 180 and 240 W power were the same indicates that the variation in power had no significant effect on the outcome of the drying process.
- Line 328: Figure 5 – please amend this because there is no Figure 5 in this manuscript.
AUTHORS: Thank you for this observation. It is corrected to Figure 4.

Round 2
Reviewer 1 Report
Comments and Suggestions for Authors
The manuscript was improved however experiment (analythical methods) was not properly designed if there were no colour analyses, texture and in general sensory parameters of obtained dried fruits...
Author Response
Dear Editor and Reviewers,
The authors extend their appreciation to the Editor and Reviewers for their prompt and expert evaluation. The Reviewers' expertise in the field is evident. All comments provided by the Reviewers have been taken into account, and the paper has been modified accordingly. The authors are confident that the revised paper meets the Reviewers' standards and is sufficiently compelling for publication in Foods. We have updated the manuscript in response to the Reviewers' feedback, clearly indicating the revisions made in the updated version. The authors supplemented the manuscript with the desired references and suggestions from the Reviewers.
Rev. 1
The manuscript was improved however experiment (analythical methods) was not properly designed if there were no colour analyses, texture and in general sensory parameters of obtained dried fruits...
Authors: The authors would like to thank the Reviewers for their helpful comments. The lack of analytical methods, such as color analyses, texture assessments, and sensory evaluations, in an experimental focus, was because the main focus was on biochemical composition (polyphenols and antioxidant capacity), drying kinetics, and energy consumption of microwave and convective drying. However, they will certainly be analyzed and included in the following research plans.

Reviewer 2 Report
Comments and Suggestions for Authors
The authors have addressed some of the issues raised by the the reviewer. However there are some questions that need improvement. I think the discussion should be improved with more refererences to other similar works.
New sentences should be improved as they are poorly written.
Ln192-194. I do not understand this sentence. As I underestand a higher content will facilitate the vaporation of water (higher MR).
Comments on the Quality of English Language
Ln175-176. Please improve this sentence "was the constant mass of dehydrated fruit".
Ln 203. DR was grown. Please improve the sentence.
Ln 205-206. Please improve the sentence. It is not easy to understand.
Ln 213-215. Please improve the sentence.
Author Response
Dear Editor and Reviewers,
The authors extend their appreciation to the Editor and Reviewers for their prompt and expert evaluation. The Reviewers' expertise in the field is evident. All comments provided by the Reviewers have been taken into account, and the paper has been modified accordingly. The authors are confident that the revised paper meets the Reviewers' standards and is sufficiently compelling for publication in Foods. We have updated the manuscript in response to the Reviewers' feedback, clearly indicating the revisions made in the updated version. The authors supplemented the manuscript with the desired references and suggestions from the Reviewers.
Rev. 2
The authors have addressed some of the issues raised by the the reviewer. However there are some questions that need improvement. I think the discussion should be improved with more refererences to other similar works.
New sentences should be improved as they are poorly written.
Ln192-194. I do not understand this sentence. As I underestand a higher content will facilitate the vaporation of water (higher MR).
Authors: The authors express gratitude for the observation. The sentence has been re-written.
“A decreased moisture ratio (MR) results in a decreased final moisture content, while an increased MR leads to a higher ultimate moisture content. Greater moisture content promotes water evaporation.”
Ln175-176. Please improve this sentence "was the constant mass of dehydrated fruit".
Authors: Thank you for the correction. The sentence has been improved.
“The drying process concluded once the dehydrated fruit reached a constant mass.”
Ln 203. DR was grown. Please improve the sentence.
Authors: The authors are grateful for the proposal. The sentence has been improved.
“As temperature and power increased, the DR also increased.”
Ln 205-206. Please improve the sentence. It is not easy to understand.
Authors: Thank you so much for your kind observation. The sentence has been improved.
“The highest DR was achieved at equivalent dehydration durations for both examined blackberry fruits, employing the identical dehydration model”
Ln 213-215. Please improve the sentence.
Authors: Thank you so much for your kind notice. The sentence has been improved.
“The similarity in results between microwave drying at power levels of 180 and 240 W indicates that the fluctuation in power did not have a notable impact on the drying outcome.”
